# Adaptive Intelligent Tracking Control of Flexible-Joint Manipulator With Full-State Constraints

1st Xinfeng Shao
*College of Science, Liaoning University of Technology*
Jinzhou, Liaoning, China
shaoxinfeng2017@163.com

2nd Yongming Li
*College of Science, Liaoning University of Technology*
Jinzhou, Liaoning, China
l_y_m_2004@163.com

*Abstract*—This paper investigates the intelligent neural adaptive trajectory tracking control problem for a flexible-joint manipulator with full-state constraints. The approach uses neural networks to approximate the system's uncertain nonlinear terms and integrates backstepping recursion with a tangent-type barrier Lyapunov function to develop an intelligent adaptive control method with comprehensive state constraints. The stability of the closed-loop system is demonstrated using Lyapunov stability theory, ensuring that all state variables remain within predetermined boundaries. The proposed method's feasibility and effectiveness are ultimately validated through simulation of the flexible-joint manipulator system.

*Index Terms*—Adaptive intelligent control, full-state constraints, backstepping, flexible-joint manipulator

## I. Introduction

THe manipulator is a vital component of industrial automation, embodying the integration of modern and automation technologies [1]. It plays a significant role in both everyday life and various production processes. Since the mid-20th century, the rapid advancement of automation and computer technologies has established a robust foundation for the development of manipulators, leading to significant progress in related research [2]. However, traditional rigid robotic arms face several limitations, including low control precision, slow response speed, and a low load-to-weight ratio, which restrict their application in high-precision manufacturing fields such as medical technology and aerospace [3]. To overcome these limitations, flexible-joint manipulators have gained considerable attention in recent years [4]. These robotic arms offer substantial advantages in terms of adaptability, safety, and precision, thereby significantly enhancing production efficiency and workplace safety [5]. There have been many notable research outcomes related to flexible-joint manipulators, with the development of an effective control system being crucial for optimizing overall performance.

The backstepping control method is highly effective for solving trajectory tracking control problems in flexible joint

This work is supported in part by the National Natural Science Foundation of China (U22A2043), the Doctoral Startup Fund of Liaoning University of Technology under Grant XB2024014.

robotic arms [6]. Many studies have utilized backstepping-based approaches to address control issues in flexible robotic arms with notable results. Huang *et al.* [7] designed a controller for a single-link flexible robotic arm based on backstepping, but this control strategy is limited to systems with known state parameters. Liu *et al.* [8] combined singular perturbation control and adaptive control with backstepping to develop a control strategy for flexible dual-robot systems, but this approach is complex to compute. To address this computational complexity, Li *et al.* [9] combined backstepping with an instruction filter and used neural networks to approximate system uncertainties, reducing computational demands. Subsequently, Dong *et al.* [10] added an error compensation mechanism to eliminate potential filtering errors introduced by the filter. Zhang *et al.* [11] applied finite-time and state-constrained backstepping methods to flexible joint robotic arms, achieving good results in handling trajectory tracking issues.

In practical production settings, there are often stringent requirements on the operational workspace of flexible-joint robotic arms [12]. Exceeding these limits can damage the system, cause shutdowns, or jeopardize operator safety. Therefore, research into flexible-joint robotic arm systems under state constraints is of paramount importance [13]. Currently, backstepping-based control strategies are prominent and show promising control performance in the study of these systems.

Motivated by the above considerations, this paper aims to investigate the adaptive NN tracking control problem for a class of flexible-joint manipulator systems with full-state constraints. Compared with the existing results, the main contributions of this paper can be summarized as:

i) Modeling flexible-joint manipulator involves using neural networks to effectively handle system nonlinearities, combined with the backstepping method to provide an adaptive intelligent tracking control approach for such systems.

ii) To ensure the stable and reliable operation of the flexible-joint manipulator, a tangent-type barrier Lyapunov function is introduced to achieve safety constraint control over system states. An intelligent adaptive full-state constraints

control method is proposed.

## II. PROBLEM FORMULATIONS AND PRELIMINARIES

### A. System Formulation and Some Assumptions

The model of the flexible joint manipulator systems considered in this paper is as follows

$$
\begin{cases}
M(q)\ddot{q} + C(q,\dot{q})\dot{q} + G(q) + F\dot{q} + K(q - q_m) = 0 \\
J\ddot{q}_m + B\dot{q}_m + K(q_m - q) = u \\
y = q
\end{cases}
\tag{1}
$$

where $q, \dot{q}, \ddot{q} \in R$ represents the manipulator displacement, velocity, and acceleration, respectively. $M(q) \in R$, $G(q) \in R$, $C(q,\dot{q}) \in R$, and $F \in R$ represents inertial matrix, Gravitational term, Coriolis centripetal matrix, and Joint friction coefficient matrix. $q_m, \dot{q}_m, \ddot{q}_m \in R$ represents the rotor angle displacement, angular velocity, and angular acceleration, respectively. $K \in R$, $J \in R$, and $B \in R$ represents the joint compliance, inertia term, and damping term of the joint, respectively. $u$ and $y$ is the system input and output, respectively.

Let $[x_1, x_2, x_3, x_4]^T = [q, \dot{q}, q_m, \dot{q}_m]^T$, the system (1) is transformed into the following form

$$
\begin{cases}
\dot{x}_1 = x_2 \\
\dot{x}_2 = f_2(\bar{x}_2) + g_2 x_3 \\
\dot{x}_3 = x_4 \\
\dot{x}_4 = f_4(\bar{x}_4) + g_4 u \\
y = x_1
\end{cases}
\tag{2}
$$

where $f_2 = -M^{-1}(x_1)\left[C(x_1, x_2)x_2 + G(x_1) + F(x_2) + Kx_1\right]$, $g_2 = M^{-1}(x_1)K$, $f_4 = -J^{-1}\left[Bx_4 + K(x_3 - x_1)\right]$, and $g_4 = J^{-1}$.

**Assumption 1** (see [14], [15]): There exists a positive constant $g_{i0}$ such that $0 < g_{i0} \leq |g_i(\bar{x}_i)| < \infty$. Without loss of generality, the assumption that $0 < g_{i0} \leq g_i(\bar{x}_i) < \infty$ holds.

**Assumption 2** (see [16], [17]): It is assumed that the reference signal $y_m(t)$ and its time derivatives $\dot{y}_m, \ddot{y}_m, y_m^{(3)}, y_m^{(4)}$ are continuous and bounded, i.e., $|y_m(t)| \leq \bar{Y}_0 < k_{c1}$ and $\left|y_m^{(i)}(t)\right| < Y_i$.

## III. ADAPTIVE INTELLIGENT FULL-STATE CONSTRAINTS CONTROL AND STABILITY ANALYSIS

### A. Adaptive Intelligent Control Design

In this section, the adaptive intelligent control design will be carried out in the backstepping recursion framework. First, define the following coordinate transformation

$$
\begin{cases}
z_1 = x_1 - y_m \\
z_i = x_i - \alpha_{i-1}, i = 2, 3, 4
\end{cases}
\tag{3}
$$

Based on the above coordinate transformation, the fuzzy adaptive backstepping recursive design process is as follows

**Step 1:** According to (2) and (3), the following equality can be obtained

$$
\begin{aligned}
\dot{z}_1 &= x_2 - \dot{y}_m \\
&= (z_2 + \alpha_1) - \dot{y}_m
\end{aligned}
\tag{4}
$$

By using neural networks $\hat{h}_1(Z_1|\hat{\theta}_1) = \hat{\theta}_1^T \varphi_1(Z_1)$ to approximate the nonlinear function $h_1(Z_1) = -\dot{y}_m$ with $Z_1 = [x_1, y_m]^T$, the following can be obtained

$$
h_1(Z_1|\theta_1) = \theta_1^{*T} \varphi_1(Z_1) + \varepsilon_1(Z_1)
\tag{5}
$$

where $\theta_1^*$ is optimal weight vector, $\varepsilon_1(Z_1)$ is NN approximation error and satisfies $|\varepsilon_1| \leq \varepsilon_1^*$ with $\varepsilon_1^* > 0$.

Construct the following $tan$-type barrier Lyapunov function

$$
V_1 = \frac{1}{2}\frac{k_{b1}^2}{\pi}\tan\left(\frac{\pi z_1^2}{2k_{b1}^2}\right) + \frac{1}{2}g_{10}\tilde{\theta}_1^2
\tag{6}
$$

where $k_{b1} = k_{c1} - \bar{Y}_0$, $\tilde{\theta}_1 = \hat{\theta}_1 - \theta_1$ is weight parameter estimation error and $\theta_1 = g_{10}^{-1}\|\theta_1^*\|^2$. From (4)-(6), we have

$$
\begin{aligned}
\dot{V}_1 &= \sec\left(\frac{\pi z_1^2}{2k_{b1}^2}\right)z_1\left[(z_2 + \alpha_1) + \theta_1^{*T}\varphi_1(Z_1) + \varepsilon_1(Z_1)\right] \\
&\quad + g_{10}\tilde{\theta}_1\dot{\hat{\theta}}_1
\end{aligned}
\tag{7}
$$

By using Young's inequality, we have

$$
\begin{aligned}
&\sec\left(\frac{\pi z_1^2}{2k_{b1}^2}\right)z_1\theta_1^{*T}\varphi_1(Z_1) \\
&\leq \frac{a_1^2}{2} + \frac{1}{2a_1^2}\sec^2\left(\frac{\pi z_1^2}{2k_{b1}^2}\right)z_1^2\|\theta_1^*\|^2\|\varphi_1(Z_1)\|^2
\end{aligned}
\tag{8}
$$

where $a_1 > 0$ is a design parameter.

$$
\begin{aligned}
&\sec\left(\frac{\pi z_1^2}{2k_{b1}^2}\right)z_1\varepsilon_1(Z_1) \\
&\leq \frac{1}{2}\sec^2\left(\frac{\pi z_1^2}{2k_{b1}^2}\right)z_1^2 g_{10} + \frac{1}{2g_{10}}\varepsilon_1^{*2}
\end{aligned}
\tag{9}
$$

Substituting (8) and (9) into (7), one has

$$
\begin{aligned}
\dot{V}_1 &\leq \sec\left(\frac{\pi z_1^2}{2k_{b1}^2}\right)z_1\left[\alpha_1 + \frac{1}{2a_1^2}\sec\left(\frac{\pi z_1^2}{2k_{b1}^2}\right)\right. \\
&\quad \left. \times z_1 g_{10}\hat{\theta}_1\|\varphi_1(Z_1)\|^2 + \frac{1}{2}\sec\left(\frac{\pi z_1^2}{2k_{b1}^2}\right)z_1 g_{10}\right] \\
&\quad + \sec\left(\frac{\pi z_1^2}{2k_{b1}^2}\right)z_1 z_2 + \frac{a_1^2}{2} + \frac{\varepsilon_1^{*2}}{2g_{10}} \\
&\quad + g_{10}\tilde{\theta}_1\left[\dot{\hat{\theta}}_1 - \frac{1}{2a_1^2}\sec^2\left(\frac{\pi z_1^2}{2k_{b1}^2}\right)z_1^2\|\varphi_1(Z_1)\|^2\right]
\end{aligned}
\tag{10}
$$

The virtual controller and the parameter adaptive update law are designed as follows

$$
\begin{aligned}
\alpha_1 &= -\lambda_1 z_1 - \frac{1}{2a_1^2}\sec\left(\frac{\pi z_1^2}{2k_{b1}^2}\right)z_1\hat{\theta}_1\|\varphi_1(Z_1)\|^2 \\
&\quad - \frac{1}{2}\sec\left(\frac{\pi z_1^2}{2k_{b1}^2}\right)z_1
\end{aligned}
\tag{11}
$$

$$
\dot{\hat{\theta}}_1 = -\sigma_1\hat{\theta}_1 + \frac{1}{2a_1^2}\sec^2\left(\frac{\pi z_1^2}{2k_{b1}^2}\right)z_1^2\|\varphi_1(Z_1)\|^2
\tag{12}
$$

where $\lambda_1, \sigma_1 > 0$ are design parameter. Substituting (11) and (12) into (10), one has

$$
\begin{aligned}
\dot{V}_1 &\leq -\sec\left(\frac{\pi z_1^2}{2k_{b1}^2}\right)z_1^2 g_{10}\lambda_1^* - \sigma_1 g_{10}\tilde{\theta}_1\hat{\theta}_1 \\
&\quad + \sec\left(\frac{\pi z_1^2}{2k_{b1}^2}\right)z_1 z_2 + \frac{a_1^2}{2} + \frac{\varepsilon_1^{*2}}{2g_{10}}
\end{aligned}
\tag{13}
$$

where $\lambda_1^* = \lambda_1 - 0.5 > 0$ is a design parameter.

**Step 2:** According to (2) and (3), the following equality can be obtained

$$\dot{z}_2 = f_2 + g_2(z_3 + \alpha_2) - \dot{\alpha}_1 \tag{14}$$

Let $h_2(Z_2) = f_2 - \dot{\alpha}_1 + \sec\left(\frac{\pi z_1^2}{2k_{b1}^2}\right) z_1 \Big/ \sec\left(\frac{\pi z_2^2}{2k_{b2}^2}\right)$ with $Z_2 = \left[\bar{x}_2, y_m, \dot{y}_m, \ddot{y}_m, \hat{\theta}_1\right]^T$. By using neural networks $\hat{h}_2(Z_2|\hat{\theta}_2) = \hat{\theta}_2^T \varphi_2(Z_2)$ to approximate the nonlinear function $h_2(Z_2)$, one has

$$h_2(Z_2|\theta_2) = \theta_2^{*T}\varphi_2(Z_2) + \varepsilon_2(Z_2) \tag{15}$$

where $\theta_2^*$ is optimal weight vector, $\varepsilon_2(Z_2)$ is NN approximation error and satisfies $|\varepsilon_2| \leq \varepsilon_2^*$ with $\varepsilon_2^* > 0$.

Construct the following $tan$-type barrier Lyapunov function

$$V_2 = V_1 + \frac{1}{2}\frac{k_{b2}^2}{\pi}\tan\left(\frac{\pi z_2^2}{2k_{b2}^2}\right) + \frac{1}{2}g_{20}\tilde{\theta}_2^2 \tag{16}$$

where $k_{b2} = k_{c2} - \bar{Y}_0$, $\tilde{\theta}_2 = \hat{\theta}_2 - \theta_2$ is weight parameter estimation error and $\theta_2 = g_{20}^{-1}\|\theta_2^*\|^2$. From (14)-(16), we have

$$\dot{V}_2 = \dot{V}_1 + \sec\left(\frac{\pi z_2^2}{2k_{b2}^2}\right) z_2 \left[g_2(z_3+\alpha_2) + \theta_2^{*T}\varphi_2(Z_2)\right.$$
$$\left. + \varepsilon_2(Z_2) - \sec\left(\frac{\pi z_1^2}{2k_{b1}^2}\right) z_1 \Big/ \sec\left(\frac{\pi z_2^2}{2k_{b2}^2}\right)\right] + g_{20}\tilde{\theta}_2\dot{\hat{\theta}}_2 \tag{17}$$

By employing Young's inequality, the following inequalities are obtained

$$\sec\left(\frac{\pi z_2^2}{2k_{b2}^2}\right) z_2\theta_2^{*T}\varphi_2(Z_2)$$
$$\leq \frac{a_2^2}{2} + \frac{1}{2a_2^2}\sec\left(\frac{\pi z_2^2}{2k_{b2}^2}\right) z_2^2\|\theta_2^*\|^2\|\varphi_2(Z_2)\|^2 \tag{18}$$

where $a_2 > 0$ is a design parameter.

$$\sec\left(\frac{\pi z_2^2}{2k_{b2}^2}\right) z_2\varepsilon_2(Z_2) \leq \frac{1}{2}\sec\left(\frac{\pi z_2^2}{2k_{b2}^2}\right) z_2^2 g_{20} + \frac{\varepsilon_2^{*2}}{2g_{20}} \tag{19}$$

From inequalities (18) and (19) yields

$$\dot{V}_2 \leq \dot{V}_1 + \sec\left(\frac{\pi z_2^2}{2k_{b2}^2}\right) z_2 \left[g_2\alpha_2 + \frac{1}{2a_2^2}\sec\left(\frac{\pi z_2^2}{2k_{b2}^2}\right)\right.$$
$$\left. \times z_2 g_{20}\hat{\theta}_2\|\varphi_2(Z_2)\|^2 + \frac{1}{2}\sec\left(\frac{\pi z_2^2}{2k_{b2}^2}\right) z_2 g_{20}\right]$$
$$+ g_{20}\tilde{\theta}_2\left[\dot{\hat{\theta}}_2 - \frac{1}{2a_2^2}\sec^2\left(\frac{\pi z_2^2}{2k_{b2}^2}\right) z_2^2\|\varphi_2(Z_2)\|^2\right] \tag{20}$$
$$- \sec\left(\frac{\pi z_1^2}{2k_{b1}^2}\right) z_1 z_2 + \sec\left(\frac{\pi z_2^2}{2k_{b2}^2}\right) z_2 z_3 g_2$$
$$+ \frac{a_2^2}{2} + \frac{\varepsilon_2^{*2}}{2g_{20}}$$

The virtual controller and the parameter adaptive update law are designed as follows

$$\alpha_2 = -\lambda_2 z_2 - \frac{1}{2a_2^2}\sec\left(\frac{\pi z_2^2}{2k_{b2}^2}\right) z_2\hat{\theta}_2\|\varphi_2(Z_2)\|^2$$
$$- \frac{1}{2}\sec\left(\frac{\pi z_2^2}{2k_{b2}^2}\right) z_2 \tag{21}$$

$$\dot{\hat{\theta}}_2 = -\sigma_2\hat{\theta}_2 + \frac{1}{2a_2^2}\sec^2\left(\frac{\pi z_2^2}{2k_{b2}^2}\right) z_2^2\|\varphi_2(Z_2)\|^2 \tag{22}$$

where $\lambda_2, \sigma_2 > 0$ are design parameter. Substituting (21) and

(22) into (20), one has

$$\dot{V}_2 \leq -\sec\left(\frac{\pi z_1^2}{2k_{b1}^2}\right) z_1^2 g_{10}\lambda_1^* - \sigma_1 g_{10}\tilde{\theta}_1\hat{\theta}_1 + \frac{a_1^2}{2}$$
$$+ \frac{\varepsilon_1^{*2}}{2g_{10}} - \sec\left(\frac{\pi z_2^2}{2k_{b2}^2}\right) z_2^2 g_{20}\lambda_2^* - \sigma_2 g_{20}\tilde{\theta}_2\hat{\theta}_2 \tag{23}$$
$$+ \frac{a_2^2}{2} + \frac{\varepsilon_2^{*2}}{2g_{20}} + \sec\left(\frac{\pi z_2^2}{2k_{b2}^2}\right) z_2 z_3 g_2$$

where $\lambda_2^* = \lambda_2 - 0.5 > 0$ is a design parameter.

**Step 3:** According to (2) and (3), we have

$$\dot{z}_3 = x_4 - \dot{\alpha}_2$$
$$= (z_4 + \alpha_3) - \dot{\alpha}_2 \tag{24}$$

Let $h_3(Z_3) = -\dot{\alpha}_2 + \sec\left(\frac{\pi z_2^2}{2k_{b2}^2}\right) z_2 g_2 \Big/ \sec\left(\frac{\pi z_3^2}{2k_{b3}^2}\right)$ with $Z_2 = \left[\bar{x}_3, y_m, \dot{y}_m, \ddot{y}_m, y_m^{(3)}, \hat{\theta}_1, \hat{\theta}_2\right]^T$. By using neural networks $\hat{h}_3(Z_3|\hat{\theta}_3) = \hat{\theta}_3^T\varphi_3(Z_3)$ to approximate the nonlinear function $h_3(Z_3)$, one has

$$h_3(Z_3|\theta_3) = \theta_3^{*T}\varphi_3(Z_3) + \varepsilon_3(Z_3) \tag{25}$$

where $\theta_3^*$ is optimal weight vector, $\varepsilon_3(Z_3)$ is NN approximation error and satisfies $|\varepsilon_3| \leq \varepsilon_3^*$ with $\varepsilon_3^* > 0$.

Construct the following $tan$-type barrier Lyapunov function

$$V_3 = V_2 + \frac{1}{2}\frac{k_{b3}^2}{\pi}\tan\left(\frac{\pi z_3^2}{2k_{b3}^2}\right) + \frac{1}{2}g_{30}\tilde{\theta}_3^2 \tag{26}$$

where $k_{b3} = k_{c3} - \bar{Y}_0$, $\tilde{\theta}_3 = \hat{\theta}_3 - \theta_3$ is weight parameter estimation error and $\theta_3 = g_{30}^{-1}\|\theta_3^*\|^2$. From (24)-(26), we have

$$\dot{V}_3 = \dot{V}_2 + \sec\left(\frac{\pi z_3^2}{2k_{b3}^2}\right) z_3 \left[(z_4 + \alpha_3)\right.$$
$$+ \theta_3^{*T}\varphi_3(Z_3) + \varepsilon_3(Z_3) \tag{27}$$
$$\left. - \sec\left(\frac{\pi z_2^2}{2k_{b2}^2}\right) z_2 \Big/ \sec\left(\frac{\pi z_3^2}{2k_{b3}^2}\right)\right] + g_{30}\tilde{\theta}_3\dot{\hat{\theta}}_3$$

By employing Young's inequality, the following inequalities are obtained

$$\sec\left(\frac{\pi z_3^2}{2k_{b3}^2}\right) z_3\theta_3^{*T}\varphi_3(Z_3)$$
$$\leq \frac{a_3^2}{2} + \frac{1}{2a_3^2}\sec\left(\frac{\pi z_3^2}{2k_{b3}^2}\right) z_3^2\|\theta_3^*\|^2\|\varphi_3(Z_3)\|^2 \tag{28}$$

where $a_3 > 0$ is a design parameter.

$$\sec\left(\frac{\pi z_3^2}{2k_{b3}^2}\right) z_3\varepsilon_3(Z_3) \leq \frac{1}{2}\sec\left(\frac{\pi z_3^2}{2k_{b3}^2}\right) z_3^2 g_{30} + \frac{\varepsilon_3^{*2}}{2g_{30}} \tag{29}$$

From inequalities (28) and (29) yields

$$\dot{V}_3 \leq \dot{V}_2 + \sec\left(\frac{\pi z_3^2}{2k_{b3}^2}\right) z_3 \left[\alpha_3 + \frac{1}{2a_3^2}\sec\left(\frac{\pi z_3^2}{2k_{b3}^2}\right)\right.$$
$$\left. \times z_3 g_{30}\hat{\theta}_3\|\varphi_3(Z_3)\|^2 + \frac{1}{2}\sec\left(\frac{\pi z_3^2}{2k_{b3}^2}\right) z_3 g_{30}\right]$$
$$+ g_{30}\tilde{\theta}_3\left[\dot{\hat{\theta}}_3 - \frac{1}{2a_3^2}\sec^2\left(\frac{\pi z_3^2}{2k_{b3}^2}\right) z_3^2\|\varphi_3(Z_3)\|^2\right] \tag{30}$$
$$- \sec\left(\frac{\pi z_2^2}{2k_{b2}^2}\right) z_2 z_3 + \sec\left(\frac{\pi z_3^2}{2k_{b3}^2}\right) z_3 z_4$$
$$+ \frac{a_3^2}{2} + \frac{\varepsilon_3^{*2}}{2g_{30}}$$

The virtual controller and the parameter adaptive update law are designed as follows

$$\alpha_3 = -\lambda_3 z_3 - \frac{1}{2a_3^2}\sec\left(\frac{\pi z_3^2}{2k_{b3}^2}\right)z_3\hat{\theta}_3\|\varphi_3(Z_3)\|^2$$
$$- \frac{1}{2}\sec\left(\frac{\pi z_3^2}{2k_{b3}^2}\right)z_3 \tag{31}$$

$$\dot{\hat{\theta}}_3 = -\sigma_3\hat{\theta}_3 + \frac{1}{2a_3^2}\sec^2\left(\frac{\pi z_3^2}{2k_{b3}^2}\right)z_3^2\|\varphi_3(Z_3)\|^2 \tag{32}$$

where $\lambda_3, \sigma_3 > 0$ are design parameter. Substituting (31) and (32) into (30), one has

$$\dot{V}_3 \le -\sum_{i=1}^{3}\sec\left(\frac{\pi z_i^2}{2k_{bi}^2}\right)z_i^2 g_{i0}\lambda_i^* - \sum_{i=1}^{3}\sigma_i g_{i0}\tilde{\theta}_i\hat{\theta}_i$$
$$+ \frac{1}{2}\sum_{i=1}^{3}a_i^2 + \frac{1}{2}\sum_{i=1}^{3}\frac{\varepsilon_i^{*2}}{g_{i0}} + \sec\left(\frac{\pi z_3^2}{2k_{b3}^2}\right)z_3 z_4 \tag{33}$$

where $\lambda_3^* = \lambda_3 - 0.5 > 0$ is a design parameter.

**Step 4:** According to (2) and (3), we have

$$\dot{z}_4 = \dot{x}_4 - \dot{\alpha}_3$$
$$= f_4 + g_4 u - \dot{\alpha}_3 \tag{34}$$

Let $h_4(Z_4) = f_4 - \dot{\alpha}_3 + \sec\left(\frac{\pi z_3^2}{2k_{b3}^2}\right)z_3\Big/\sec\left(\frac{\pi z_4^2}{2k_{b4}^2}\right)$ with $Z_4 = \left[\bar{x}_4, y_m, \ldots, y_m^{(4)}, \hat{\theta}_1, \ldots, \hat{\theta}_3\right]^T$. By using neural networks $\hat{h}_4(Z_4|\hat{\theta}_4) = \hat{\theta}_4^T\varphi_4(Z_4)$ to approximate the nonlinear function $h_4(Z_4)$, one has

$$h_4(Z_4|\theta_4) = \theta_4^{*T}\varphi_4(Z_4) + \varepsilon_4(Z_4) \tag{35}$$

where $\theta_4^*$ is optimal weight vector, $\varepsilon_4(Z_4)$ is NN approximation error and satisfies $|\varepsilon_4| \le \varepsilon_4^*$ with $\varepsilon_4^* > 0$.

Construct the following $tan$-type barrier Lyapunov function

$$V_4 = V_3 + \frac{1}{2}\frac{k_{b4}^2}{\pi}\tan\left(\frac{\pi z_4^2}{2k_{b4}^2}\right) + \frac{1}{2}g_{40}\tilde{\theta}_4^2 \tag{36}$$

where $k_{b4} = k_{c4} - \bar{Y}_0$, $\tilde{\theta}_4 = \hat{\theta}_4 - \theta_4$ is weight parameter estimation error and $\theta_4 = g_{40}^{-1}\|\theta_4^*\|^2$. From (24)-(26), we have

$$\dot{V}_4 = \dot{V}_3 + \sec\left(\frac{\pi z_4^2}{2k_{b4}^2}\right)z_4\big[g_4 u$$
$$+ \theta_4^{*T}\varphi_4(Z_4) + \varepsilon_4(Z_4) \tag{37}$$
$$- \sec\left(\frac{\pi z_3^2}{2k_{b3}^2}\right)z_3\Big/\sec\left(\frac{\pi z_4^2}{2k_{b4}^2}\right)\big] + g_{40}\tilde{\theta}_4\dot{\hat{\theta}}_4$$

By employing Young's inequality, the following inequalities are obtained

$$\sec\left(\frac{\pi z_4^2}{2k_{b4}^2}\right)z_4\theta_4^{*T}\varphi_4(Z_4)$$
$$\le \frac{a_4^2}{2} + \frac{1}{2a_4^2}\sec\left(\frac{\pi z_4^2}{2k_{b4}^2}\right)z_4^2\|\theta_4^*\|^2\|\varphi_4(Z_4)\|^2 \tag{38}$$

where $a_4 > 0$ is a design parameter.

$$\sec\left(\frac{\pi z_4^2}{2k_{b4}^2}\right)z_4\varepsilon_4(Z_4) \le \frac{1}{2}\sec\left(\frac{\pi z_4^2}{2k_{b4}^2}\right)z_4^2 g_{40} + \frac{\varepsilon_4^{*2}}{2g_{40}} \tag{39}$$

From inequalities (38) and (39) yields

$$\dot{V}_4 \le \dot{V}_3 + \sec\left(\frac{\pi z_4^2}{2k_{b4}^2}\right)z_4\Big[g_4 u + \frac{1}{2a_4^2}\sec\left(\frac{\pi z_4^2}{2k_{b4}^2}\right)$$
$$\times z_4 g_{40}\hat{\theta}_4\|\varphi_4(Z_4)\|^2 + \frac{1}{2}\sec\left(\frac{\pi z_4^2}{2k_{b4}^2}\right)z_4 g_{40}\Big]$$
$$+ g_{40}\tilde{\theta}_4\Big[\dot{\hat{\theta}}_4 - \frac{1}{2a_4^2}\sec^2\left(\frac{\pi z_4^2}{2k_{b4}^2}\right)z_4^2\|\varphi_4(Z_4)\|^2\Big] \tag{40}$$
$$- \sec\left(\frac{\pi z_3^2}{2k_{b3}^2}\right)z_3 z_4 + \frac{a_4^2}{2} + \frac{\varepsilon_4^{*2}}{2g_{40}}$$

The final adaptive controller and parameter adaptive update law are designed as follows

$$u = -\lambda_4 z_4 - \frac{1}{2a_4^2}\sec\left(\frac{\pi z_4^2}{2k_{b4}^2}\right)z_4\hat{\theta}_4\|\varphi_4(Z_4)\|^2$$
$$- \frac{1}{2}\sec\left(\frac{\pi z_4^2}{2k_{b4}^2}\right)z_4 \tag{41}$$

$$\dot{\hat{\theta}}_4 = \frac{1}{2a_4^2}\sec^2\left(\frac{\pi z_4^2}{2k_{b4}^2}\right)z_4^2\|\varphi_4(Z_4)\|^2 - \sigma_4\hat{\theta}_4 \tag{42}$$

where $\lambda_4, \sigma_4 > 0$ are design parameter.

### B. Stability Analysis

The above control design is summarized as the following theorem.

**Theorem 1**: For the single-link flexible joint manipulator systems (1), Assumptions 1 and 2 hold. If the virtual controllers are given by (11), (21), and (31), the controller is designed by (41), and the adaptive law is designed by (42), the designed intelligent adaptive control scheme can ensure the following performance objectives:

a) All signals in the closed loop system are SGUUB.
b) The tracking error converges to a smaller neighborhood containing the origin.
c) All the states of the system satisfy the constraint boundaries given in advance.

**Proof**: Substituting (41) and (42) into (40), one has

$$\dot{V}_4 \le -\sum_{j=1}^{4}\sec\left(\frac{\pi z_j^2}{2k_{bj}^2}\right)z_j^2 g_{j0}\lambda_j^* - \frac{1}{2}\sum_{j=1}^{4}\sigma_j g_{j0}\tilde{\theta}_j\hat{\theta}_j$$
$$+ \frac{1}{2}\sum_{j=1}^{4}a_j^2 + \frac{1}{2}\sum_{j=1}^{4}\frac{\varepsilon_j^{*2}}{g_{j0}} \tag{43}$$

where $\lambda_4^* = \lambda_4 - 0.5 > 0$ is a design parameter. By employing Young's inequality, we have

$$-\sigma_j g_{j0}\tilde{\theta}_j\hat{\theta}_j \le -\frac{1}{2}\sigma_j g_{j0}\tilde{\theta}_j^2 + \frac{1}{2}\sigma_j g_{j0}\theta_j^2 \tag{44}$$

By substituting (44) into (43), it can be obtained

$$\dot{V}_4 \le -\sum_{j=1}^{4}\sec\left(\frac{\pi z_j^2}{2k_{bj}^2}\right)z_j^2 g_{j0}\lambda_j^* - \frac{1}{2}\sum_{j=1}^{4}\sigma_j g_{j0}\tilde{\theta}_j^2$$
$$+ \frac{1}{2}\sum_{j=1}^{4}\frac{a_j^2}{2} + \frac{1}{2}\sum_{j=1}^{4}\frac{\varepsilon_j^{*2}}{2g_{j0}} + \frac{1}{2}\sum_{j=1}^{4}\sigma_j g_{j0}\theta_j^2 \tag{45}$$

For $|z_j| < k_{bj}$, $\frac{k_{bj}^2}{\pi} \tan\left(\frac{\pi z_j^2}{2k_{bj}^2}\right) < \sec\left(\frac{\pi z_j^2}{2k_{bj}^2}\right) z_j^2$ holds, thus

$$\dot{V}_4 \leq - \sum_{j=1}^{4} \frac{k_{bj}^2}{\pi} \tan\left(\frac{\pi z_j^2}{2k_{bj}^2}\right) g_{j0}\lambda_j^*$$
$$- \frac{1}{2} \sum_{j=1}^{4} \sigma_j g_{j0} \tilde{\theta}_j^2 + \Xi \tag{46}$$

where $\Xi = \frac{1}{2} \sum_{j=1}^{4} \frac{a_j^2}{2} + \frac{1}{2} \sum_{j=1}^{4} \frac{\varepsilon_j^{*2}}{2g_{j0}} + \frac{1}{2} \sum_{j=1}^{4} \sigma_j g_{j0}\theta_j^2$. Choose $\beta = \min\left\{2g_{j0}\lambda_j^*, \sigma_j, j = 1,2,3,4\right\}$, we have

$$\dot{V}_4 \leq -\beta V_4 + \Xi \tag{47}$$

Similar to [18]-[20], the closed-loop signals are bounded. In addition, from $z_1 = y - y_m$, $|y_m(t)| \leq \bar{Y}_0$, $|x_1| \leq |z_1| + |y_m| \leq k_{b1} + \bar{Y}_0$, we have $|x_1| \leq k_{c1}$. Moreover, since $\alpha_{j-1}(\cdot)(j = 2,3,4)$ are some bounded continuous functions of bounded signals, we have $|\alpha_{j-1}(\cdot)| \leq \bar{\alpha}_{j-1}$. According to $x_j = z_j + \alpha_{j-1}$ and $|z_j| \leq k_{bj}(j = 2,3,4)$, we have $|x_j| \leq |z_j| + \bar{\alpha}_{j-1} = k_{cj}$. Therefore, all states of the system do not violate its pregiven constraint bounds. ∎

## IV. SIMULATION RESEARCH

In this part, in order to further verify the rationality and effectiveness of the developed control design method, the following single-link flexible manipulator simulation example is provided.

In the simulation, the desired reference trajectory at the end of the link is chosen as $y_m(t) = \sin(0.5t) + 0.5\sin(t)$. The constraint boundary for the four states is chosen as $k_{cj} = 2$ $(j = 1,2,3,4)$. The parameters of the single-link flexible manipulator model are given in TABLE 1.

**TABLE I**

PARAMETERS OF THE MANIPULATOR SYSTEM

| Description | Values | Unit |
|---|---|---|
| The length-$L$ of link | 1.0 | $m$ |
| The link mass $M$ | 0.3 | $kg$ |
| The natural damping $B$ | 0.015 | none |
| The actuator inertia $J$ | 0.35 | $kg * m^2$ |
| The joint flexible $K$ | 15 | $N \cdot m/rad$ |

The simulation results are shown in Figs. 1-7. Fig. 1 shows the tracking curve of the system output, and Fig. 2 shows the tracking error curve. It can be seen that the tracking error is very small and converges to within a small neighborhood of zero. Figs. 3-6 show the curves of four states and constraint boundaries, and it can be seen that all states remain within the preset constraint boundaries. Fig. 7 shows the system control input curve, which verifies the effectiveness of the proposed adaptive intelligent full-state constraints control method.

## V. CONCLUSION

This paper addresses the problem of adaptive intelligent full-state constraints control for single-link flexible manipulator

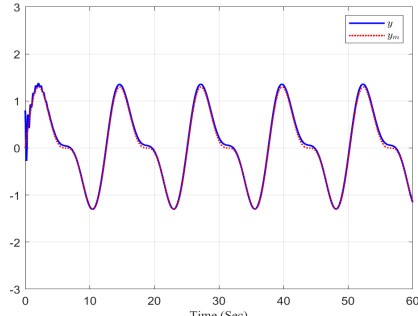

Fig. 1.   Curves of output $y$ and desired signal $y_m$.

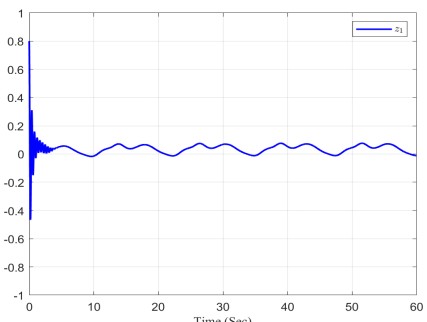

Fig. 2.   Curve of the tracking error $z_1$.

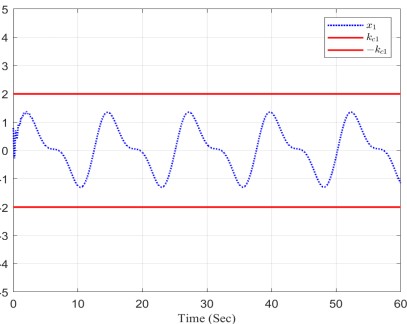

Fig. 3.   Curves of the state $x_1$ and the constraint boundary $k_{c1}$.

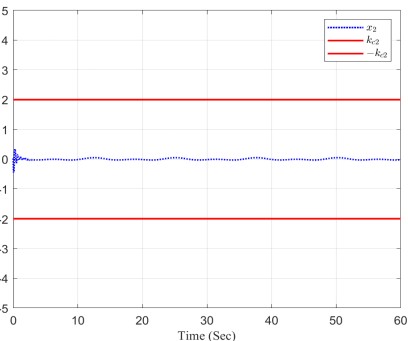

Fig. 4.   Curves of the state $x_2$ and the constraint boundary $k_{c2}$.

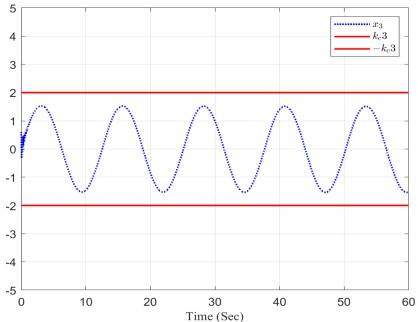

Fig. 5. Curves of the state $x_3$ and the constraint boundary $k_{c3}$.

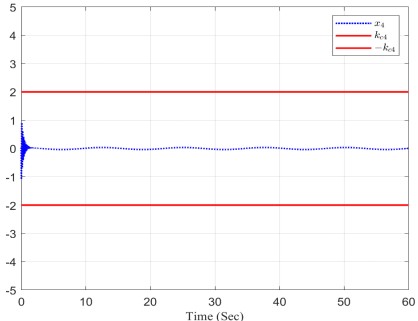

Fig. 6. Curves of the state $x_4$ and the constraint boundary $k_{c4}$.

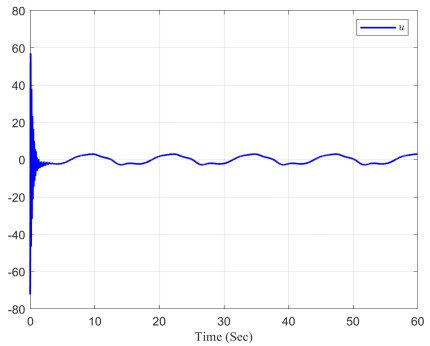

Fig. 7. Curve of the controller $u$.

systems. Utilizing neural network approximation techniques effectively handles the uncertain nonlinearities in the systems. An intelligent adaptive neural network controller has been proposed. By combining a novel barrier Lyapunov function method, a controller with a full-state constraint structure is designed, ensuring the stability of the closed-loop system. Finally, the proposed method is applied to a single-link flexible manipulator system, demonstrating its effectiveness.

## ACKNOWLEDGMENT

This work is supported in part by the National Natural Science Foundation of China (U22A2043), the Doctoral Startup Fund of Liaoning University of Technology under Grant XB2024014.

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
