# OpenReview forum: "Adaptive Intelligent Tracking Control of Flexible-Joint Manipulator With Full-State Constraints"
_IEEE.org/ICIST/2024/Conference — IEEE ICIST 2024 Conference Submission_

### Official Review · Reviewer_Y5DH · 2024-08-21
**This paper is very interesting and a good work.**

**Rating:** 10
**Confidence:** 5

**Review:**

This paper investigates the intelligent neural adaptive trajectory tracking control problem for a flexible-joint manipulator with full-state constraints. The theory is correct and can be accepted after responding the following comments.
(1)	In the introduction, the shortages of those relevant studies are suggested to be further summarized.
(2)	In the end of Section 1, the organization of this study is suggested to be summarized.
(3)	In equation 2, check the notation “${f_2}$” and “${f_4}$”. Should they be replaced by ${f_2}({\bar x_2})$ and ${f_4}({\bar x_4})$?
(4)    The future work is missing in the Conclusion.

---

### Official Review · Reviewer_BV1V · 2024-08-21
**This paper investigates the intelligent neural adaptive trajectory tracking control problem for a flexible-joint manipulator with full-state constraints. The approach uses neural networks to approximate the system's uncertain nonlinear terms and integrates backstepping recursion with a tangent-type barrier Lyapunov function to develop an intelligent adaptive control method with comprehensive state constraints. The obtained result is valuable and can be accepted if the following problems can be clarified.**

**Rating:** 7
**Confidence:** 3

**Review:**

1. This study may have different designs in comparison to some previous studies, but how are these differences significant? The author needs to highlight and clarify them in innovation point.
2. Some remarks should be added to make the reader understand this paper well.
3. The direction of the next work should be explained in the conclusion.

---

### Official Review · Reviewer_d42D · 2024-08-22
**Accept**

**Rating:** 8
**Confidence:** 5

**Review:**

This paper studied the adaptive intelligent tracking control for manipulator with full state constarins. The following comments should be considered in the final version.
A. The control objective of this paper should be added.
B. The definition of SGUUB should be added.
C. The authors should use the present perfect tense in the conclusion.

---

### Official Review · Reviewer_QVad · 2024-08-23
**Adaptive Intelligent Tracking Control of Flexible-Joint Manipulator With Full-State Constraints**

**Rating:** 7
**Confidence:** 2

**Review:**

This paper investigates the intelligent neural adaptive trajectory tracking control problem for a flexible-joint manipulator with full-state constraints. The topic of this paper is interesting, but the writing and the organization of this paper needs to be improved. Some comments are provided as follows:
1.	For the fluency of reading, language should be expressed consistently before and after. For example, tangent-type, flexible-joint manipulator, etc. A careful rereading is recommended.
2.	There are still many grammar errors and typos in this paper. For example, in the explanation section following (5), (6), the article is missing before a noun, a/the should be added, or the plural form should be used; in the introduction, "The" or "THE" is better than "THe".
3.	The abbreviated form should be given at the first occurrence in the article, e.g. NN.

---

### Official Review · Reviewer_Pheb · 2024-08-23
**Accept,  but some suggestions remain**

**Rating:** 6
**Confidence:** 5

**Review:**

This paper introduces a tangent-type barrier Lyapunov function and  the state constraints method to achieve the intelligent neural adaptive trajectory tracking control for a flexible-joint manipulator. The derivation of this paper is adequate, but some suggestions remain:
1. Image analysis of the simulation could be more refined.
2. Grammatical descriptions could be more standardised.

---

### Decision · Program_Chairs · 2024-09-08

Accept (Oral)